



# The relation between crystal structure
# and the occurrence of quantum-rotor induced polarization
Corinna Dietrich,[1] Julia Wissel,[1] Oliver Lorenz,[1] Arafat Hossain Khan,[2] Marko Bertmer,[2] Somayeh Khazaei,[3]
Daniel Sebastiani,[3] Jörg Matysik[1]
[1]Institut für Analytische Chemie, Universität Leipzig, Linnéstr. 3, 04103 Leipzig, Germany
[2]Felix-Bloch-Institut für Festkörperphysik, Universität Leipzig, Linnéstr. 5, 04103 Leipzig, Germany
[3]Institut für Chemie, Martin-Luther-Universität Halle-Wittenberg, Von-Danckelmann-Platz 4, 06120 Halle,
Germany
**Correspondance:** Jörg Matysik (joerg.matysik@uni-leipzig.de)
ABSTRACT
Among hyperpolarization techniques, quantum-rotor induced polarization (QRIP), also known as Haupt effect,
is a peculiar one. It is on one hand rather simple to apply by cooling and heating of a sample. On the other
hand, only the methyl groups of a few substances seem to allow for the effect, which strongly limits the ap-
plicability of QRIP. While it is known, that a high tunnel frequency is favorable, the structural requirements for
the effect to occur are not exhaustively studied yet. Here we report on our efforts to heuristically recognize
structural motifs in molecular crystals able to allow to produce QRIP.



## 1 INTRODUCTION

NMR spectroscopy is a very versatile analytical method, however, caused by the low Boltzmann ratio, suffers from a lack of sensitivity. Therefore, hyperpolarization methods are presently a "hot" issue (Halse, 2016; Köckenberger and Matysik, 2010; Kovtunov et al., 2018; Wang et al., 2019). Examples of these techniques are dynamic nuclear polarization (Ardenkjaer-Larsen, 2016; Kjeldsen et al., 2018; Lilly Thankamony et al., 2017; Milani et al., 2015; Ni et al., 2013), spin-exchange optical pumping (Hollenbach et al., 2016; Meersmann and Brunner, 2015; Norquay et al., 2018; Walker, 2011), photochemically induced dynamic nuclear polarization (Bode et al., 2013; Kiryutin et al., 2012; Sosnovsky et al., 2019) and para-hydrogen induced polarization

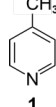

**Figure 1:** $\gamma$-picoline (**1**).

(Duckett and Mewis, 2012; Kiryutin et al., 2017; Korchak et al., 2009). Another technique is quantum-rotor induced polarization (QRIP) (Dumez et al., 2017; Horsewill, 1999; Icker et al., 2013; Icker and Berger, 2012; Ludwig et al., 2010). It was first observed by Haupt in $\gamma$-picoline (**1**, Figure 1) during rapid temperature jumps at very low temperatures (Haupt, 1972, 1973).

It is also possible to access the signal enhancement in liquid state, by freezing **1** at helium temperature and then rapidly dissolving it in deuterated solvents at room temperature and measuring it immediately (Icker and Berger, 2012). With a custom-made setup, we were able to improve the safety and speed of the dissolution and transfer process, resulting in a higher signal enhancement factor of 530 (Dietrich et al., 2018). An example of a QRIP enhanced spectrum is given in Figure 2. The enhancement is limited to the signal of the methyl carbon and exhibits a before unexpected antiphase pattern (Icker and Berger, 2012).

One might expect that more methyl bearing compounds allow for QRIP, which would broaden the applicability of the effect. However, work of Icker and Berger (Icker et al., 2013) indicated that only a few substances with methyl groups can be hyperpolarized in this way and all of the positively tested compounds show a weaker hyperpolarization compared to **1**. Therefore, the structural requirements for the occurrence of QRIP need to be elucidated.



For a deeper understanding of these requirements, we discuss the underlying mechanisms of the effect. Ther-
modynamically, QRIP has been interpreted in terms of a resonant contact between a tunnelling reservoir and
a Zeeman reservoir (Horsewill, 1999) at low temperatures. The nuclear spin-order is produced via coupling of
spin-states to rotational quantum states of the methyl group. At the temperature of liquid helium only the lowest
rotational state is occupied. Upon a dramatic temperature jump to room temperature, a measurable non-Boltz-
mann distribution of nuclear spin states is gained via cross-relaxation effects. These relaxations also explain
the antiphase pattern and are further described in (Roy et al., 2013).

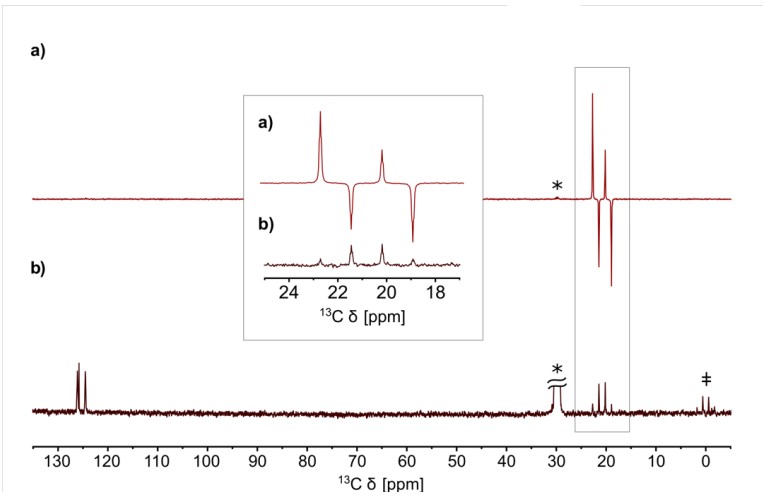

**Figure 2:** a) QRIP enhanced $^{13}C$ spectrum of $\gamma$-picoline (**1**) measured without proton decoupling, recorded with one scan after the cooling and dissolution procedure. Acetone-d6 was used as solvent. b) Reference spectrum recorded after full relaxation with 100 scans. The signals of acetone-d6 (labelled with asterix, *) and TMS (‡) are strongly visible in the reference spectrum.

The energy gap between the rotational ground state and the first excited state determines the (low-temperature)
population ratio via the Boltzmann factor and thus the overall amplitude of the imposed spin symmetry con-
straint. Thus, it can be expected that high tunnel frequencies are strongly favorable for observing QRIP effects.
In fact, **1** has an exceptionally high tunnel frequency of 520 $\mu$eV (~ 4 cm$^{-1}$) (Prager and Heidemann, 2010).
The tunnel frequency is also linked to the capability of the methyl group to rotate freely (Barlow et al., 1992).
Therefore, structural motifs with free methyl groups are especially interesting. In the case of **1**, the crystal
structure shows a rather special feature. Each methyl group is paired with another one and the pairs are all
aligned perfectly in a face-to-face manner. Around both methyl pairs, the chemical environment creates rota-
tional potential energy barriers (often with a $C_3$ or $C_6$ symmetry). There is a strong coupling of both these
methyl groups (due to their spatial proximity) with a $2\pi/6$ phase difference, which means that the superposition
of the two rotational potential energy functions becomes surprisingly flat (i.e. the hills of the first rotational
potential just fit to the valleys of the second potential function). This in turn leads to the possibility for joint
rotation of the methyl groups at very low rotational barrier, virtually a free rotation, eventually resulting in a very
high tunnel splitting (Khazaei and Sebastiani, 2017).
In the present work, we therefore search for substances which have one or several of these features: methyl
groups with low steric hindrance, methyl groups in a similar distance to each other and face-to-face arrange-
ment as in **1**, and methyl groups with concerted rotations.





2 MATERIALS AND METHODS
*2.1 Liquid-state NMR*
Experiments were carried out on a Bruker Fourier-300 and a Bruker DRX-400 spectrometer. For the QRIP
studies, samples were cooled for 90 min in liquid helium and subsequently mixed with deuterated solvents at
room temperature. The mixture was transferred to the magnet and measured immediately. This procedure
was carried out manually or with the self-built transfer system where the mixing and the transfer of the solution
into the magnet is carried out in one step during 35 s (Dietrich et al., 2018). If suitable, the transfer system has
been preferred, due to faster sample transfer into the magnet. In cases of insufficient solubility, solely the
manual procedure has been found to be applicable. To validate structures and determine the signal enhance-
ment factor, reference spectra were measured after full relaxation of the enhancement. Therefore, multiple
scans were recorded, whereas QRIP enhanced spectra have been obtained with a single scan.
*2.2 Solid-state NMR*
For the solid-state experiments under magic-angle-spinning (MAS), a Bruker Avance III spectrometer (400
MHz $^1$H frequency) was used. In order to test for QRIP enhancement, the powder sample was packed into a
4-mm zirconia rotor, closed with a zirconia cap and cooled for 90 min in liquid helium. After cooling, the rotor
was transferred manually into the magnet and spectra were recorded. For the measurement under vacuum,
the powder sample was filled into a glass tube (3 mm outer diameter) and evacuated over 2 days. Afterwards,
the glass tube was sealed and fitted into the 4-mm zirconia rotor with polytetrafluoroethylene stoppers (Khan
et al., 2018). The rotor was closed with a zirconia cap and used as before. In every case, non-decoupled Hahn
echo pulse sequences were used and the spinning frequency was set to 8 kHz. Again, reference spectra with
multiple scans were recorded afterwards.
*2.3 Signal enhancement factor*
To compare QRIP enhanced spectra with one scan to reference spectra with multiple scans, the enhancement
factor $\varepsilon$ has been calculated by using equation 1. $(S/N)_{QRIP}$ is the signal to noise ratio of the QRIP-enhanced
signal and $(S/N)_{ref}$ is the signal to noise ratio of the reference spectrum with multiple scans. The number of
scans is given as $n_{ref}$ (Dietrich et al., 2018). $S/N$ ratios were obtained from the Topspin 3.1 software.

$$\varepsilon = \frac{\sqrt{n_{ref}} \cdot (S/N)_{QRIP}}{(S/N)_{ref}} \tag{1}$$


*2.4 Inelastic neutron scattering*
Inelastic neutron scattering (INS) measurements were carried out at the TOF-TOF instrument at the For-
schungs-Neutronenquelle Heinz Maier-Leibnitz (Garching, Technical University of Munich).
*2.5 X-ray diffraction*
For the powder X-ray diffraction (PXRD) patterns, the samples were placed in 0.5 mm Ø capillaries and meas-
ured using a STOE STADI P diffractometer (Cu Kα$_1$ radiation; equipped with a MYTHEN (DECTRIS) detector).
Measurements were carried out at the Institute of Inorganic Chemistry, University of Leipzig.
*2.6 Synthesis*
$\gamma$-Picoline hydrochloride (**2**) was commercially available (Carbosynth Limited), while $\gamma$-picoline nitrate (**3**) and
$\gamma$-picoline hydrosulfate (**4**) were synthesized according to instructions from (Wang et al., 2015) and (Ullah et
al., 2015).



## 1 3 EXPERIMENTAL RESULTS AND DISCUSSION

### 2 *3.1 Chemical analogues of γ-picoline*

To gain a better understanding of the conditions for the occurrence of the effect, this heuristic study aims at
finding connections between the various structural properties of a substance and the observed signal enhance-
ment by QRIP. First, molecules that are similar to **1** in their molecular structure were searched and as a result
very close analogues, three different salts of **1**, were found (Figure 3).

$$
\begin{array}{ll}
CH_3 & \\
& Cl \quad (\mathbf{2}) \\
A^{\ominus} \quad A = & NO_3 \quad (\mathbf{3}) \\
& HSO_4 \quad (\mathbf{4})
\end{array}
$$

**Figure 3:** γ-picoline derivatives **2**-**4**.

All salts are solids at room temperature and are soluble in $H_2O$. Hence, $D_2O$ was used as solvent for the liquid-
state NMR experiments. For the QRIP experiments, the manual transfer was chosen, since the high viscosity
of $D_2O$ hinders the liquid flow in the transfer system and results in air bubbles in the NMR tube inside the
magnet which will lead to a disturbed signal. From each salt, 50 $\mu$mol were cooled for 90 min at 4.2 K, quickly
dissolved in $D_2O$, the solution (inside the NMR tube) was shortly held in an ultrasonic bath to remove air
bubbles and then transferred and measured. Even though the chemical structure and especially the chemical
environment of the methyl group seem similar to **1**, no QRIP enhancement was observed for any of the three
salts **2** to **4** in the $^{13}C$ NMR spectra. The $^{13}C$ NMR reference spectra are similar to the one of **1**, with slight
chemical shift changes (see Table 1).
**Table 1:** $^{13}C$ NMR reference spectra of γ-picoline and its derivatives recorded on a Bruker Fourier-300 spectrometer. $D_2O$ was used as
solvent. The chemical shifts are given in ppm.

| substance | assignment | | | |
|---|---|---|---|---|
| | **C**—$CH_3$ | N=**CH**—CH | CH—**CH**=$C_q$ | **$CH_3$** |
| γ-picoline (**1**) | 152.5 | 150.9 | 128.0 | 23.0 |
| γ-picoline hydrochloride (**2**) | 161.7 | 140.0 | 127.9 | 21.9 |
| γ-picoline nitrate (**3**) | 164.4 | 142.8 | 130.6 | 24.5 |
| γ-picoline hydrosulfate (**4**) | 164.5 | 142.8 | 130.7 | 24.5 |

Remarkably, also other chemical analogues, like the *α*-form and the *β*-form of picoline, show no signal en-
hancement as was shown already before in the work of M. Icker et al. (Icker et al., 2013). They also have
studied toluene (**5**), which is in its chemical structure very similar to **1** and shows little QRIP enhancement.
Hence, we conclude that not the molecular structure is decisive for successful induction of QRIP and that
already small modifications of the molecular structure can decide upon either QRIP induction or quenching.



Searching for other parameters controlling the occurrence of QRIP, we recognize that lithium acetate dihydrate
(**6**), which is no picoline analogue, shows moderate QRIP enhancement (weaker than **1**, stronger than **5**).
Comparing the crystal structures, we found that both **1** and **6** exhibit pairs of methyl groups facing each other
in a 180° angle (Figure 4 a and c), crystal structures from (Galigné et al., 1970; Ohms et al., 1985)), while the

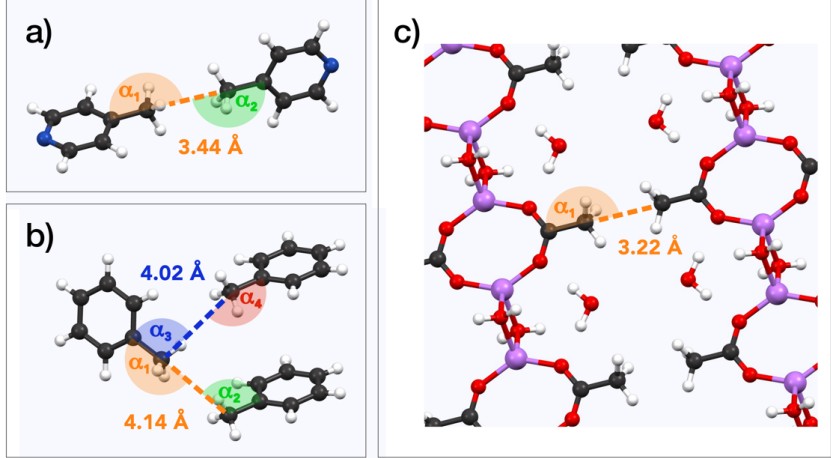

**Figure 4:** Examples for methyl pairs in the crystal structure. The distance between methyl pairs is given by the carbon-to-carbon distance. Angles are measured along the path from quaternary carbon via methyl carbon to the methyl carbon of the neighbor molecule. a) $\gamma$-Picoline (**1**) methyl pairs from crystal structure ZZZIVG, angles $\alpha_1$ and $\alpha_2$ vary between 177° and 180° (Ohms et al., 1985). b) Toluene (**5**) methyl pairs (TOLUEN), the distance between the closest pairs is either 4.02 Å or 4.14 Å. Respective angles: $\alpha_1 = 165°$, $\alpha_2 = 97°$, $\alpha_3 = 94°$, $\alpha_4 = 157°$ (van der Putten et al., 1990). c) Methyl pairs of lithium acetate dihydrate (**6**, LIACET), the angle $\alpha_1$ is 180° (Galigné et al., 1970). The ionic bonds (Ac$^-$ ⋯ Li$^+$ ⋯ OH$^2$) are plotted the same as regular covalent bonds in order to improve the spatial comprehensibility of the crystal representation.

methyl groups in **5** have no such symmetry (Figure 4 b), crystal structure from (van der Putten et al., 1990)).
This might explain the different tunnel frequencies, which directly affect QRIP (see Table 2) (Icker et al., 2013;
Roy et al., 2013). Since there is hardly "empty" space in condensed phase, in most crystal structures methyl
groups cannot rotate freely. Only the direct compensation of two rotational barriers of two methyl groups which
show a 180° face-to-face arrangement allows for almost "frictionless" rotations of the coupled methyls ("con-
certed" rotations, (Khazaei and Sebastiani, 2017)). Therefore, such a spatial arrangement in the crystal might
provide a rare but well-defined structural feature allowing for induction of QRIP.
**Table 2:** Comparison of structural properties and QRIP: methyl-methyl- (Me-Me-) distances were measured carbon to carbon, angles
between methyl groups were measured along the path from quaternary carbon via methyl carbon to the methyl carbon of the neighbor
molecule (received from crystal structure data (Faber et al., 1999; Galigné et al., 1970; Ohms et al., 1985; van der Putten et al., 1990));
tunnel frequencies from (Prager and Heidemann, 2010) and QRIP signal enhancement factor $\varepsilon$ from (Icker et al., 2013). Additionally to
the name of the substance, the crystal structure code is given in parentheses.

| substance (structure code) | Me-Me-distance [Å] | Me-Me-angle | tunnel frequency [$\mu$eV (cm$^{-1}$)] | QRIP $\varepsilon$ |
|---|---|---|---|---|
| $\gamma$-picoline (**1**, ZZZIVG) | 3.44 | 177°-180° | 520 ($\sim$ 4) | 60 |
| $\gamma$-picoline hydrochloride (**2**, DICCEX) | 6.31 | 99° | - | - |
| toluene (**5**, TOLUEN) | 4.02/4.14 | 94°-165° | 28.5/26.0 ($\sim$ 0.2) | 3 |
| lithium acetate dihydrate (**6**, LIACET) | 3.22 | 180° | 250 ($\sim$ 2) | 20 |



The present work aims for further corroborating the experimental evidence of this correlation. Hence, our next
step has been the systematic search for substances, which have structural properties similar to γ-picoline (**1**)
in regards to the methyl-methyl distance and the face-to-face arrangement of the methyl groups. To this end,
we searched for compounds of matching crystal structures.
### 3.2 Systematic crystal structure search
To find promising candidates for QRIP signal enhancement, the Cambridge Crystallography Database was
searched for substances with similar distances and angles between methyl groups compared to those values
given in Table 2. Other desired properties were relatively small molecular size (to have a high methyl concen-
tration and better chances to observe signal) and commercial availability. All six selected substances are listed
in Table 3.
**Table 3:** investigated compounds **7-12** from the systematic crystal structure search. Methyl-methyl- (Me-Me-) distances were measured
carbon to carbon, angles between methyl groups were measured along the path from quaternary carbon via methyl carbon to the methyl
carbon of the neighbor molecule. The crystal structure data was obtained from the Cambridge Crystallography Database. Additionally to
the name of the substance, the crystal structure CODE is given in parentheses.

| substance (structure code) | molecular structure | Me-Me-distance [Å] | Me-Me-angle |
|---|---|---|---|
| *N*-(*p*-tolyl)acetamide (**7**, ACTOLD) | | 3.61 | 153°/170° |
| 2,5-dimethyl-1,3-dinitrobenzene (**8**, AYOYAP) | | 3.55 | 168° |
| N-(*tert*-butyl)acetamide (**9**, APUYIU) | | 3.58 | 161° |
| Ethyl carbamate (**10**, ECARBM) | | 3.53 | 171° |
| 2-nitropropane (**11**, IHIKIV) | | 3.23/3.46 | 90°-150° |
| *N'*-(3,4-difluorobenzylidene)-4-me-thylbenzenesulfonohydrazide (**12**, NUQDUA) | | 3.60 | 172° |





Although distances and angles between methyl groups of those compounds are in the range between com-
pounds **1** and **5**, none of these compounds show a perfect face-to-face alignment of the methyl groups, and
no QRIP enhancement was observed for any of them. The most probable reason is the occurrence of steric
hindrance around the methyl groups in the crystal packing. This might affect the free rotation of the methyl

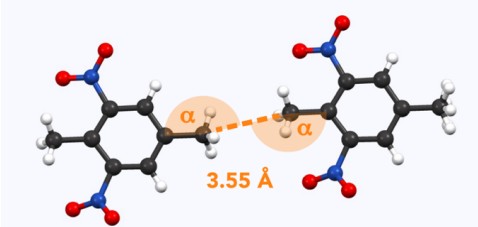

**Figure 5:** Example for methyl pairs in the crystal structure of **8**
(AYOYAP) (Johnston and Crather, 2011). The distance between
them is 3.55 Å (measured carbon to carbon). Respective angles
(measured carbon to carbon to carbon): α = 168°.

groups, and, thus lead to lower tunnel frequencies inhibiting QRIP. Despite similar angles and distances of
methyl groups, the impact of steric effects of the whole structure is difficult to estimate. To validate this corre-
lation, theoretical calculations as in (Khazaei and Sebastiani, 2016, 2017) and experimental measurements of
tunnelling frequencies are desirable. Another limitation can be a low concentration of methyl groups. In case

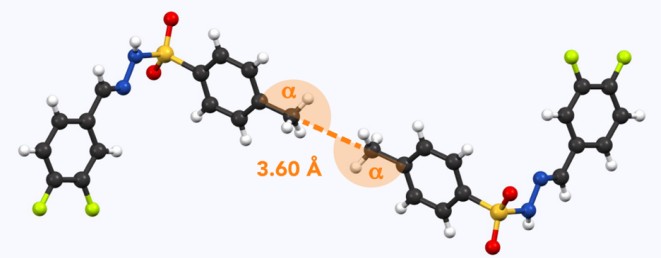

**Figure 6:** Example for methyl pairs in the crystal structure of **12** (NUQDUA) (Wang
and Yan, 2015). The distance between them is 3.60 Å (measured carbon to carbon).
Respective angles (measured carbon to carbon to carbon): α = 172°.

of low QRIP (as exhibited in **5**), higher amounts of the sample were necessary to observe a QRIP enhanced
signal, i.e. 150 $\mu$mol for a good signal, whereas for **1** 50 $\mu$mol are sufficient to observe an intense QRIP en-
hanced signal. For compounds **7** to **11**, depending on the solubility between 50 and 100 $\mu$mol substance were
used and in the case of **12** only 30 $\mu$mol was suitable.



Compounds **8** and **12** were further investigated, since they were the 2 most promising candidates of this series.
Interestingly, their methyl groups are in almost perfect face-to-face alignment (see Figure 5 (Johnston and
Crather, 2011) and Figure 6 (Wang and Yan, 2015)). On the other hand, they differ in the alignment of the
attached phenyl rings compared to **1**. While the phenyl rings of two molecules lie in the same plane for **8** and
**12**, they are tilted 90° to each other in case of compound **1** (Figure 4 a)). Whether this structural difference has
an impact on QRIP requires further theoretical investigations. To rule out that the obtained substances are
amorphous or possess another crystal structure as compared to the literature, we performed X-ray diffraction
(XRD) and confirmed the correct crystal structure of compounds **8** and **12** (Figure 7 and Figure 8).

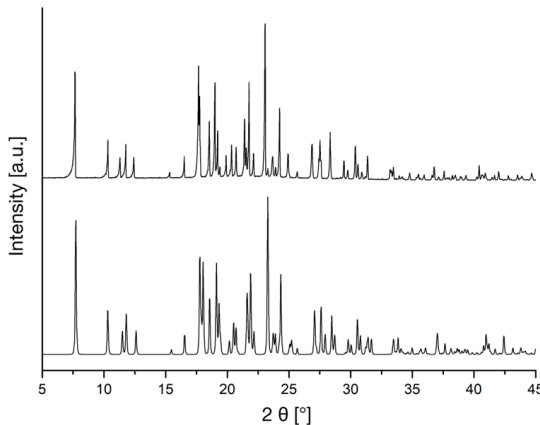

**Figure 7:** XRD spectra of **8**. Top: experimental data , bottom: simulated spectrum from crystal structure data  (Johnston and Crather, 2011).

Furthermore, the tunnel frequency has been investigated by inelastic neutron scattering (INS). A multi-peak fit
allowed to determine the first two transitions for each compound. For compound **8** we determined values of
40±10 $\mu$eV and 150±10  $\mu$eV (0.3 cm$^{-1}$ and 1.2 cm$^{-1}$). For compound **12** we found 50±10  $\mu$eV and 160±10 $\mu$eV

**Figure 8:** XRD spectra of **12**. Top: experimental data , bottom: simulated spectrum from crystal structure data (Wang and Yan, 2015).

(0.4 cm$^{-1}$ and 1.3 cm$^{-1}$). These values lie in the range between the tunnel frequencies of **5** and **6** (26/28 $\mu$eV
and 250 $\mu$eV), which both exhibit QRIP enhancement, but to a lower extent than **1**. Therefore, in regards to
the tunnel frequencies, QRIP in compounds **8** and **12** is conceivable but very likely to exhibit only a weak
enhancement.





It is noteworthy, that in **8** each molecule possesses 2 methyl groups, thus the methyl concentration is higher
compared to tests on **1** and is not expected to be the limitation. On the other hand, steric hindrance due to the
$NO_2$ groups in close proximity to the methyl group might limit QRIP. In the case of **12**, there is no such hin-
drance through intramolecular factors, however, intermolecular hindrance is conceivable and the low concen-
tration (30 $\mu$mol) is the most probable limitation.

### 3.3 Aspirin

Next to the crystallographic databank approach, we searched for compounds having a particularly low fre-
quency mode of the methyl group. In its crystal structure, aspirin (acetylsalicylic acid, **13**) has a particularly low
frequency mode near 30 cm$^{-1}$ (3.7 meV), attributed to the concerted motions of methyl groups (Reilly and
Tkatchenko, 2014). Compound **13** became an object of interest, since it exhibits some similarity to the com-
pound **1**, in which the collective coupled motions of methyl groups are contributing to QRIP and the calcu-
lated methyl rotational barrier height of **1** is about 3.57 me (Khazaei and Sebastiani, 2016). Analysing the
crystal structure of **13**, we found no face-to-face methyl pairs (ACSALA (Arputharaj et al., 2012)). The closest
methyl pairs are in a distance of 4.43 Å to each other and the angles between them are 100°/147°. Multiple
dissolution experiments showed no QRIP enhancement. According to (Prager and Heidemann, 2010), the
tunnel frequency of **13** is 1.22 $\mu$eV (0.01 cm$^{-1}$), which is much lower than the tunnel frequencies of **1** and **2**
(see Table 2). In fact, the mere coupling between two methyl groups (be it via a face-to-face arrangement or
via lateral coupling similar to a cogwheel couple) is not sufficient for allowing a free rotation (leading to high
tunnel splittings). A mandatory additional condition is that the rotational barriers created by the crystal sur-
roundings have just the correct offset to each other. Assuming the common $C_n$ symmetries for the rotational
barriers, this means that the maximum of the rotational potential for one of the methyl groups has to coincide
exactly with the minimum of the rotational potential of the other one. This additional condition seems to be not
fulfilled for aspirin, leading to the absence of QRIP enhancement.

### 3.4 Calixaren complexes

Furthermore, we considered two types of compounds following our "chemical intuition": calixarene compounds
and metal-organic frameworks. Calixarenes can occur in a cone shape and are therefore able to host smaller
molecules like toluene (Gutsche, 1981). Because of the highly symmetric structure inside the calixarene cone,
we suspected a favourable situation for the methyl group of the guest toluene molecule to rotate freely. Thus,
there might be a possibility to observe QRIP enhancement in this complex. Hence, we tested two calixarenes
as hosts: calix[4]arene (**14**) and 4-*t*-butylcalix[4]arene (**15**) (see Figure 9).

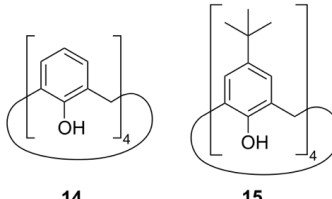

**14**        **15**

**Figure 9:** Structures of calix[4]arene (**14**) and 4-
t-butylcalix[4]arene (**15**).

Complexes toluene@calix[4]arene (**16**) and toluene@4-*t*-butylcalix[4]arene (**17**) were synthesized by mixing
a surplus of toluene with each calixarene at room temperature and letting the excess liquid dry (Andreetti et
al., 1979).
In both cases, we did not succeed to obtain a sufficiently high concentration in solution in order to perform
QRIP experiments. This is due to the weak solubility of the calixarene complexes in CDCl$_3$, acetone-d6 and
toluene-d8, often resulting in opaque solutions or white suspensions with precipitate even for low concentra-
tions. For compound **16** the solubility is higher than for **17**. In the best case, we achieved an almost clear
solution of 20 mg of **16** in CDCl$_3$. Due to the higher mass of the complex in comparison to **1**, the resulting
concentration is below what we expect to be observable by means of QRIP with the current setup. In future



studies, calixarene complexes might be studied by solid-state NMR, avoiding the solubility issue. Furthermore,
complexes with rather soluble calixarenes (Rehm et al., 2009) might provide an opportunity.
***3.5 Metal Organic Frameworks (MOFs)***
Compared to molecular crystals, MOFs provide an alternative approach to observe freely rotating methyl
groups. Methyl groups with low steric hinderance are, for example, expected in MOFs such as ZIF-8 and
ZIF-67 (zeolitic imidazole framework, see Figure 10). The difference in these two compounds lies in the differ-
ent metal centre atoms: Zn(II) in ZIF-8 and Co(II) in ZIF-67. Due to the specific structure allowing for pore
formation, the methyl groups are pointing toward the center of these pores and thus can rotate freely, which
has been shown at cryogenic temperatures (Li et al., 2018; Zhou et al., 2008).

**Figure 10:** Structure of probed MOFs: ZIF-8 and ZIF-67.

Due to the low solubility and in order to avoid hindrance of free rotating methyl groups by solvent molecules
inside the pores, the measurements were carried out in solid-state NMR. To ensure that carbon signal of both
samples can be observed in general, reference spectra were recorded before and after the QRIP experiments.
Examples of these spectra are given in Figure 11. The assignments of the signals are given in Table 4.
**Table 4:** $^{13}$C NMR reference spectra of ZIFs recorded on a Bruker Avance III spectrometer (400 MHz $^1$H frequency, solid-state NMR).
The chemical shifts are given in ppm.

| substance | assignment | | |
|---|---|---|---|
| | N—**C**(CH$_3$)=N | N—**C**H=**C**H—N | **C**H$_3$ |
| ZIF-8 | 143.4 | 128-110 | 17-7 |
| ZIF-67 | 143.3 | 130-110 | 17-7 |






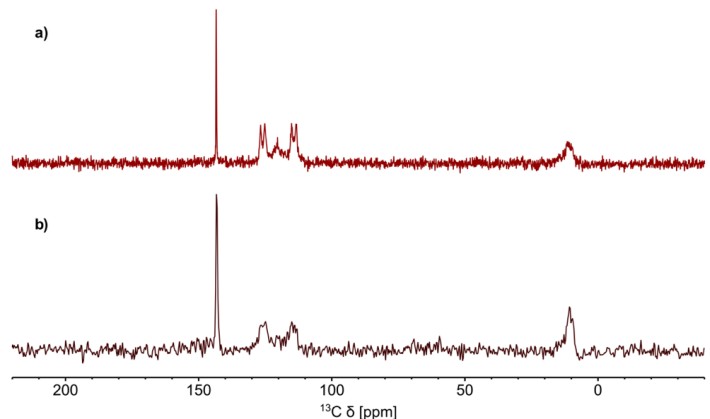

**Figure 11:** a) Spectrum of ZIF-8 measured in glass tube inside rotor. Hahn echo
pulse sequence, non-decoupled, 8 kHz MAS frequency, 10.000 scans.
b) Spectrum of ZIF-67 measured with regular packing method.
Hahn echo pulse sequence, non-decoupled, 8 kHz MAS frequency, 1.000 scans.

In both cases, $^{13}$C QRIP experiments were performed under MAS-conditions. The resulting spectra showed
no signal enhancement. A possible explanation for the absence of QRIP can be the adsorption of air molecules
inside the pores of the ZIFs (Moggach et al., 2011), which might hinder the free rotation of the methyl groups.
To exclude this, subsequent measurements under vacuum were performed. Also in this case, both ZIF sam-
ples did not show QRIP.
It is noteworthy, that Co(II) is paramagnetic and hence, signal broadening and otherwise unexpected chemical
shifts (Bertini et al., 2005; Gueron, 1975; Vega and Fiat, 1976) were expected. However, the obtained spec-
trum (Figure 11 b) shows no significant difference in the chemical shift in comparison to the Zn(II) analog
(Figure 11 a). The signals are slightly broadened. Furthermore, an especially narrow line (20 Hz for ZIF-8, and
74 Hz for ZIF-67) at the far left of the spectrum is observed. The $^{13}$C MAS NMR spectra were recorded without
proton-decoupling since decoupling is expected to interfere with the QRIP effect. Therefore, CH and CH$_3$ sig-
nals of the ZIFs are broad, while the quaternary carbon is less affected. For the latter the closest proton is the
one from the methyl group with a distance between the quarternary carbon and the methyl proton of 2.14 Å
for ZIF-8 and 2.06 Å for ZIF-67. From this distance we calculated a CH-dipole-dipole coupling of 3083 Hz (ZIF-
8), 3456 Hz (ZIF.67), which is averaged out at the chosen spinning speed of 8 kHz. With this and the high
structural symmetry, resulting in a low chemical shift anisotropy, the narrow line can be explained.
For both ZIFs reference spectra with reasonable signal intensity were recorded after 1.000 scans for the reg-
ular packing method and 10.000 scans for the advanced packing method with glass tubes. Following equation
1 the signal enhancement factor $\varepsilon$ should be at least 32 (regular packing, or 100 for advanced packing) in order
to observe signal with one scan. Smaller signal enhancement via QRIP is conceivable but could not be ob-
served with the current setup. An intrinsic limitation to QRIP might be the proximity between the methyl groups
inside the pores (ZIF-8: 5.0 Å, crystal structure data from (Morris et al., 2012); ZIF-76: 4.6 Å (Kwon et al., 2015);
measured from carbon to carbon). This might also lead to the absence of signal. Considering the broad variety
of MOFs, it is conceivable, that some of them bear methyl groups, which rotate more freely or undergo con-
certed rotations, which are accessible for QRIP (Gangu et al., 2016; Gonzalez-Nelson et al., 2019; Kuc et al.,
2007; Tarasi et al., 2020; Tian et al., 2007).

### 3.6 Analysis of previous data
Aiming for heuristic data on the relation between structure and the hyperpolarization obtained by QRIP, we
revisited the crystal structures of compounds studied in (Icker et al., 2013) and (Icker, 2013). From all sub-
stances which were available in the Cambridge Crystallography Database, angles and distances between
methyl pairs were extracted. Here, we searched specifically for methyl pairs with a similar distance as found



in **1** (3.44 Å) and excluded all methyl pairs with a distance > 4.4 Å. The results are given in Table 5. If available,
the tunnel frequencies (Prager and Heidemann, 2010) were included to Table 5 as well. Interestingly, many
methyl pairs with a distance in the range 3.45-4.37 Å were found, which is quite similar to **1** and makes methyl
coupling conceivable. On the other hand, no other face-to-face methyl groups were found. Angles close to 180°
do not seem to be a sufficient argument to predict QRIP enhancement, as the comparison of two of the com-
pounds shows: while **24** exhibits an 174° angle at a methyl-methyl distance of 4.06 Å it yields only a week
polarization. Furthermore, a rather strong QRIP effect is observed in **25** where the most promising methyl pair
has similar distance (3.78 Å), however, at the same time, it is less aligned with angles of 142°/158°. The
surprisingly high polarization in **25** is still below **1** but larger than in **6** which is particularly curious since both **1**
and **6** exhibit face-to-face methyl groups, while **25** does not. Although the structure of **25** does not fit our
assumptions to gain QRIP, the tunnel frequency is surprisingly high, which fits the presence of QRIP.
It is possible, that the occurrence of multiple methyl groups in one molecule and multiple methyl pairs in the
crystal structure are favourable for the likelihood of concerted rotations. However, those structural factors alone
are also not sufficient for the prediction of QRIP, as other not or less polarizable substances like **7-9** (multiple
methyl groups) contradict a general trend.
**Table 5:** List of compounds tested for QRIP in (Icker et al., 2013) and (Icker, 2013). Methyl methyl distances and angles between me-
thyl groups were measured from the crystal structure data. Tunnel frequencies were taken from (Prager and Heidemann, 2010). Addi-
tionally to the name of the substance, the crystal structure CODE is given in parentheses.

| substance (structure code) | Me-Me-distance [Å] | Me-Me-angle | tunnel frequency [$\mu$eV (cm$^{-1}$)] | QRIP $\varepsilon$ |
|---|---|---|---|---|
| sodium acetate (**18**, BOPKOG) | 4.24 3.45 | 123°/142° 90° | 1.5 (~ 0.01) | low |
| acetonitrile (**19**, QQQCIV) | 3.95 | 139° | - | low |
| acetone (**20**, HIXHIF) | 3.76 3.91 | 133°/176° 132°/158° | 0.4 (~ 0.003) | low |
| $\alpha$-picoline (**21**, ZZZHKQ) | 4.09 | 63°/152° | - | 0 |
| p-xylene (**22**, ZZZITY) | 3.71 4.14 | 90°/160° 99° | 0.97 (~ 0.008) | 0 |
| p-cresol (**23**, CRESOL) | 4.01 | 95°/113° | - | 0 |
| m-cresol (**24**, MCRSOL) | 3.99 4.06 3.89 3.94 | 79°/176° 174° 83° 83°/114° | - | low |
| 1,3-dibromo-2,4,6-trime-thylbenzene (**25**, EJEROA) | 3.77 3.78 4.08 3.74 4.03 | 134° 142°/158° 129°/156° 80°/83° 99° | 390 (~ 3.1) | 28 |
| 2-methoxynaphthalene (**26**, SAYRIT) | 3.60 4.05 4.21 | 66°/172° 73°/107° 125° | - | 0 |
| 2,6-di-t-butylnaphthalene (**27**, KOKQUW) | 3.75 3.95 4.28 | 155°/161° 109°/124° 106°/136° | - | 0 |
| cholesterol (**28**, CHOEST) | 4.37 4.31 4.31 3.56 | 98°/138° 91°/148° 89°/115° 84°/108° | - | 0 |




4 CONCLUSIONS
The aim of this study was to gain further understanding of the structural requirements of substances allowing
for QRIP signal enhancement in NMR spectroscopy. Starting from the well-studied compound **1** we found that
its derivatives (**2-4**) do not exhibit QRIP. This indicates that structural similarity on a molecular level is insuffi-
cient for QRIP prediction. The weak polarization in **5** and absence of QRIP in $\alpha$-picoline and $\beta$-picoline (Icker
et al., 2013) support this lack of correlation.
To better understand the specialty of **1**, we studied the crystal structure and recognized a rare structural feature:
in its crystal structure pairs of methyl groups are aligned in a perfect 180° face-to-face manner. For the under-
lying tunnel effects freely rotating methyl groups and high tunnel frequencies are favorable. Via concerted
rotations the face-to-face methyl groups in **1** can rotate exceptionally frictionless, like interacting gear wheels
(Khazaei and Sebastiani, 2017; Roy et al., 2013).
In order to investigate the predictability and applicability of QRIP, we therefore searched for substances which
show one or multiple of the aforementioned qualities: free rotation, promising alignment, high tunnel frequency
of the methyl group, or concerted rotations of methyl groups. Thus, different approaches were tested.
First, we searched for compounds with similar methyl-methyl distances and angles as in **1** and found sub-
stances **7-12**. While all of them exhibit similar distances between methyl groups, they have no face-to-face
arrangement of methyl groups and showed no QRIP enhancement. We conclude that either steric hindrance
or missing positive interference of the methyl group is quenching the effect due to the less favorable arrange-
ment.
Next, aspirin (**13**) was tested since it is described to have concerted motions of methyl groups. However, no
QRIP enhancement was observed. We conclude that concerted rotations alone are insufficient. An additional
condition is that the rotational barriers created by the crystal surroundings have just the correct offset to each
other. This means that the maximum of the rotational potential for one of the methyl groups has to coincide
exactly with the minimum of the rotational potential of the other one.
We further suspect freely rotating methyl groups in complexes of toluene in calixarene cones and in MOFs.
While the free rotation in calixarene complexes derives from a very symmetric surrounding of the methyl group
the MOFs show methyl groups in a relatively empty space. To this end, we did not succeed to perform QRIP
measurements on calixarene complexes, due to its low solubility. In MOFs we did not observe QRIP enhance-
ment.
Finally, we revisited previously studied compounds from (Icker et al., 2013) and compared QRIP enhancement
to methyl-methyl distances and angles. In the analyzed crystal structures of **18-28** we found no face-to-face
methyl groups, but a variety of angles and distances between methyl pairs. However, no general trend or
correlation between distances/angles and the enhancement factor was found. On the contrary, we found that
**25** shows a higher polarization than **6**, despite the missing face-to-face arrangement. Although we were not
able to recognize structural patterns in the crystal structures related to the appearance of QRIP, we confirm
that a high tunnel barrier is required to induce QRIP.
To explain why promising candidates like **8** and **12** showed no QRIP and **6** exhibits weaker QRIP than **1** (both
show face-to-face methyl groups only with a slight difference in the methyl-methyl distance) we conclude that
similarly as in **13** the necessary offset between rotational barriers of the methyl group is not given and thus
QRIP is quenched. To summarize we find with this study that even small structural differences can quench the
QRIP effect by strongly affecting the tunnel frequency. Thus, a broader applicability of the effect on, for exam-
ple, protein methyl groups is not to be expected.





**Data availability.** NMR spectra were originally recorded with TopSpin and processed with MestraNova. The
TopSpin files include the raw data as well as the pulse sequences. Those files and the XRD data (Origin files)
are available from zenodo.org with https://doi.org/10.5281/zenodo.5078040.
**Supplement.** The Supplement contains the following information: spectra of $\gamma$-picoline derivatives, crystal
structure of $\gamma$-picoline hydrochloride and pictures of the glass tubes for MAS-NMR under air exclusion. The
supplement related to this article is available online at: xxxxxx.
**Author contributions.** JM and DS designed the research. JW synthesized the $\gamma$-picoline derivatives. AHK
prepared the glass tube samples for MAS-NMR under air exclusion. CD, JW and OL carried out the NMR
measurements and CD, JW, OL, JM and DS interpreted the data. The paper was written with contributions
from all the authors. All authors approved the final version of the paper.
**Competing interests.** The authors declare that they have no conflict of interest.
**Special issue statement.** This article is part of the special issue "Jeffery Bodenhausen Festschrift". It is not
associated with a conference.
**Acknowledgements.** The authors thank Prof. Dr. Stefan Berger, Dr. Maik Icker and Dr. Matthias Findeisen
(Leipzig University) for helpful discussions. We also thank Dr. Astrid Schneidewind and Dr. Wiebke Lohstroh
(Ludwig Maximilian University of Munich) for the INS measurements, and Oliver Erhart (Leipzig University) for
the XRD measurements. Furthermore, we thank Dr. Michael Ruggiero (University of Vermont) for providing us
with the ZIF-8 and ZIF-67 samples and Prof. Berthold Kersting und Dr. Peter Hahn (Leipzig University) for
providing the calixarene samples.
**Financial support.** This work was supported by the Deutsche Forschungsgemeinschaft (DFG) under Grant
number MA 4972/5-1.

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
