# Peer review of "The relation between crystal structure"

_Magnetic Resonance, 2021_

## Author Response (AR2)

**Reply to RC1:** 'Comment on mr-2021-51', Malcolm Levitt, 24 Jul 2021

*"Although the results are largely negative, I do think that this study deserves to be in the scientific literature, since it demonstrates that either gamma-picoline and the few other compounds that show a strong QRIP possess some very subtle structural feature that has escaped detection by the authors in their detailed study, or possibly that the crystal structure is not the determining factor after all. For example some subtlety of the phonon spectrum might be responsible, although I confess that I have not much of an idea where to look. Nevertheless, I do suggest that in their conclusions, the authors might at least speculate on the possibility that molecular and crystal structures are not the determining factor for this phenomenon after all.*
*In summary this is a worthwhile study, and should be published, even though …"*

We are glad to learn that the referee finds our work publishable although our results are largely negative. We also thank for the suggestion to add a speculation into the final paragraph whether the discussion of the crystal structure is sufficient to understand the phenomenon of QRIP.

*"A few small things should be corrected. It is not quite true that "only the methyl groups of a few substances seem to allow for the effect". Very weak QRIP effects have also been observed in 17O water-endofullerene (doi.org/10.1103/PhysRevLett.120.266001). "*

Ref added.

*"The authors cite Ludwig et al. (PNAS, 2010) as having studied QRIP, but the attribution of the described effects to QRIP have been disputed (doi.org/10.1016/j.jmr.2017.12.009). "*
We excluded the reference by Ludwig et al. 2010 and instead added the more suitable review by Meier 2018.

*"I was surprised to see that the article cited as Roy 2013 has a completely incorrect list of authors. That error suggests that all references should be rechecked carefully. "*
We corrected that Ref and rechecked all other refs.

*"A compilation of the studied molecular systems in one place would be helpful. In some cases one has to trawl through the text to find what a certain number refers to. "*
A table with all the investigated compounds has been added to the begin of the results chapter.

*"I do not feel that providing the X-ray structural data of some of the compounds is worthwhile in the main text (figures 7, 8). The MAS spectra of the MOFs also do not seem worthy of display in the main manuscript, especially since the QRIP results were negative. "*
We shifted these Figs to the Supplement.

*"On the other hand, the authors cite neutron scattering data which shows a tunneling splitting, but never provide this data at all. Personally, I would be more interested in seeing that."*
The data are given in the Supplement.

**Reply to RC2:** 'Comment on mr-2021-51', Benno Meier, 27 Jul 2021

*While this represents a negative finding, it is still a rather conclusive one, and suited for publication in Magnetic Resonance.*
We are glad that also this reviewer considers our ms worth to be published.

*I have the following suggestions for the manuscript:*
*- In the abstract, the authors write that "a high tunnel frequency is favorable". This should in my view be rephrased to "is required". After all, if the tunnel frequency is small, there is no quantum rotation, and - consequentially - no QRIP will be observed.*
Thanks, we changed that statement accordingly.

*- It may be confusing to readers who are not very familiar with the effect, that a free rotor shows a large tunneling splitting. Indeed, in the limit of free rotation, there should be no tunneling. It would therefore be valuable to point out that the tunneling splitting is defined as the difference between the first two rotational states, and it is the population differences across these states that give rise to QRIP.*
We added that statements into the introduction.

*- The authors find a small, but significant tunneling splitting in compounds 8 and 12, but do not explicilty report details on their attempt to observe QRIP in this compound. Was such an attempt made? What was the concentration after dissolution? Is 13C labelling possible? Perhaps it is worth pointing out, that larger molecules will also tend to "loose" quantum-rotor-induced polarization more quickly due to their longer correlation times.*
We indeed did perform three dissolution experiments on compound 8 and two dissolution experiments on compound 12. According to your suggestion we added some more details to the text and included the hint at the longer correlation times. The possibility of 13C labelling was added to the conclusion.

*- The authors report on MAS QRIP experiments. How have these actually been conducted? QRIP requires equilibration at 4 Kelvin even in the most favourable cases. Have the authors performed such a temperature jump experiment with MAS? If yes, it would be prudent to give the details of temperature vs. time. A key parameter would be, after all, the time required to ramp the temperature from 4 K to say 30 K. If no such experiment has been conducted, the MAS data will be completely inconclusive with respect to QRIP, and should not be shown in the manuscript.*
The experimental details are given in chapter 2.2. We added the information about the temperature (4.2 K helium temperature to room temperature in the magnet).

*Finally, the manuscript is very well written overall, but the jubilee's spelling in the special issue statement should be checked.*
We corrected the spelling of the name of the jubilee.

**Reply to RC3:** 'Comment on mr-2021-51', Alexej Jerschow, 31 Jul 2021

*This manuscript describes a search for quantum-rotor induced polarization effects in methyl groups of compounds similar to gamma picoline. The similarity is defined by having similar methyl-methyl distances and bond angle configurations. A number of identified compounds have been studied and the QRIP effect is either nonexistent or weak in them. This leads the authors to the conclusion that the molecular similarity criteria used are not sufficient to ensure the presence of this effect, and/or that nuances of the crystal structure are important for it to occur. The work therefore represents a negative result with regard to the initial hypothesis, while it does not preclude other factors and molecules being identified as displaying the QRIP effect in the future.*
We agree with that summery.

*A minor comment would be that clarity could be improved with regard to how narrow the initial hypothesis is. For example, is the concerted methyl rotation really the best way to insure the lowest amount of friction? Perhpas the authors may also wish to state that their study would indicate that it is not the case?*
We agree that also mechanisms other than pairwise concerted rotation of two face-to-face methyl groups are not the only and possibly not even the best way to lower the rotational barrier of methyl rotation in a crystalline environment. Further candidates include a gear-like coupling of two adjacent methyl groups (which, however, we did not observe in any real molecule), and phonon modes of the molecular crystal, which could couple to the rotational motion of a methyl. We also added that statement into the final discussion.

*Furthermore, I wonder whether relaxation effects could be of relevance here (e.g. through latttice dynamics / vibrations).*
That is a good point: Cross relaxation is an essential part of the QRIP theory as proposed by Levitt et al. Competing relaxation pathways would quench the effect. We added a short remark into the final discussion.

**Reply to Editor Review:** 'Comment on mr-2021-51', Jean-Nicolas Dumez, 30 Aug 2021

*However, one of the sentence added during revisions appears to be ambiguous:*
*p13, l36: "We also recognize, that mechanisms other than pairwise concerted rotation of*
*two face-to-face methyl groups are not the only and possibly not even the best way to*
*lower the rotational barrier of methyl rotation in a crystalline environment."*
*Could you please reformulate this sentence ?*

We reformulated the sentence to be unambiguous: "We also recognize that the mechanism of pairwise concerted rotation of two face-to-face methyl groups is not the only, and possibly not even the best, way to lower the rotational barrier of methyl rotation in a crystalline environment.".